# Oral Health Interventions in Patients with a Mental Health Disorder: A Scoping Review with Critical Appraisal of the Literature

**DOI:** 10.3390/ijerph18158113

**Published:** 2021-07-30

**Authors:** Sonja Kuipers, Nynke Boonstra, Linda Kronenberg, Annette Keuning-Plantinga, Stynke Castelein

**Affiliations:** 1Research Group Healthcare & Innovation in Psychiatry, Department of Healthcare, NHL Stenden University of Applied Sciences, Rengerslaan 8-10, 8900 CG Leeuwarden, The Netherlands; Nynke.Boonstra@nhlstenden.com (N.B.); annette.keuning@nhlstenden.com (A.K.-P.); 2Department of Clinical Psychology and Experimental Psychopathology, University of Groningen, Grote Kruisstraat 2/1, 9712 TS Groningen, The Netherlands; s.castelein@lentis.nl; 3KieN VIP, Oosterkade 72, 8911 KJ Leeuwarden, The Netherlands; 4Dimence Mental Health Care, Burgemeester Roelenweg 9, 8021 EV Zwolle, The Netherlands; L.Kronenberg@dimencegroep.nl; 5Health Sciences-Nursing Science & Education, University Medical Centre Groningen, Hanzeplein 1, P.O. Box 30.001, 9700 RB Groningen, The Netherlands; 6Lentis Research, Lentis Psychiatric Institute, Hereweg 80, 9725 AG Groningen, The Netherlands

**Keywords:** oral health, prevention, nursing, quality of life, mental disorders, psychiatric nursing

## Abstract

Poor oral health affects quality of life and daily functioning in the general population and especially in patients with mental health disorders. Due to the high burden of oral health-related quality of life in patients with a mental health disorder, it is important for nurses to know how they can intervene in an early phase. The aim of this systematic scoping review was to identify and appraise oral health interventions in patients with a mental health disorder. A systematic scoping review with a critical appraisal of the literature was conducted using the Joanna Briggs Institute (JBI) methodology for scoping reviews and their checklists. MEDLINE, CINAHL, PsycINFO and reference lists were searched from their inception until December 2020. Results: Eleven quantitative studies were included in the review: four randomized controlled trials, six quasi-experimental studies and one cohort study. Studies focused on interventions for patients (*n* = 8) or focused on patients together with their professionals (*n* = 3). Four types of oral health interventions in mental health were found: (I) educational interventions; (II) physical interventions; (III) interventions combining behavioural and educational elements and (IV) interventions combining educational and physical elements. All studies (*n* = 11) had an evaluation period ≤12 months. Nine studies showed an effect on the short term (≤12 months) with regard to oral health knowledge, oral health behaviour, or physical oral health outcomes (e.g., plaque index). Two studies showed no effects on any outcome. Overall, the methodological insufficient to good. Conclusion: Four types of interventions with positive effects (≤12 months) on oral health knowledge, oral health behaviour, and physical oral health outcomes in different diagnostic patient groups were found. Due to the heterogeneity in both interventions, diagnostic groups and outcomes, one golden standard oral health intervention cannot be advised yet, although the methodological quality of studies seems sufficient. Developing an integrated oral health toolkit might be of great importance in mental health considering its potential effect on oral health-related quality of life.

## 1. Introduction

The World Health Organisation (WHO) emphasises that oral health is integral and essential to general health and wellbeing [1,2]. Oral health is improved in the general population, but vulnerable patients (e.g., patients diagnosed with mental health disorders) have not benefited from the worldwide improvement in oral health [3]. Poor oral health is associated with diabetes (both type 1 and 2) [4], respiratory disease and abdominal obesity. It might also be related to cardiovascular diseases [4,5,6,7], but cigarette smoking might influence this relationship [8].

Nearly 20% of the population worldwide suffers from a mental health disorder [9,10], and this outlines the importance of oral health in patients diagnosed with a mental health disorder [11] who are exposed to more oral health risk factors [3,11,12,13,14].

Several risk factors of poor oral health in patients with a mental health disorder were described [15]. Many patients consume medication such as antipsychotics, antidepressants, and lithium. A dry mouth [16,17] is a side-effect of the medication which can increase plaque [18,19,20,21]. Next, oral health will be worsened by the consumption of sugary sweets and sugary drinks [22], which are more frequently used in patients with a mental health disorder. Inadequate oral health self-management, a lower tooth brushing frequency, a lack of motivation for proper oral hygiene and health care habits and poor psychosocial functioning are known as other barriers for adequate oral health in patients diagnosed with a mental health disorder [15,16,17,23]. Bad breath (halitosis) may lead to poor self-image, low self-esteem, decreased self-confidence, social phobia, loneliness, depression and suicidal intents in the general population [24,25].

Thus, poor oral health affects quality of life and daily functioning in the general population and especially in patients with a mental health disorder [1,2]. As a consequence, patients living with severe mental illness (SMI) (e.g., schizophrenia or related psychotic disorders, bipolar disorder,) are almost three times more likely to have lost all of their teeth compared to the general population [26].

It is evident that routine and effective oral care is necessary for maintaining oral health of in- and outpatients [27]. Mental health professionals (e.g., nurses) have an important role in the care for (out)patients with a mental health disorder. Therefore, nurses should consider oral health care as an essential part of their care for patients with mental health disorders [28].

Until now, existing NICE-guidelines primarily focus on oral health in general practice [29] and on adults in care homes [30]. No NICE-guideline focusses on oral health interventions of patients diagnosed with a mental health disorder, their oral health needs and risk factors (e.g., the use of antipsychotic medication). A British guideline titled “Oral Health Care for People with Mental Health Problems” [16] describes the severity and prevalence of oral health problems in mental health. This guideline does not meet the needs with outdated literature. The evidence of interventions is mostly focussed on institutionalised elderly and not on patients with mental health disorders. It is important to outline interventions in groups of mental health disorder due to the differences in management (e.g., the management of oral health of a patient with depression might differ from that of a patient with severe cognitive problems).

Considering the poor oral health, increased risk factors, the high burden of poor oral health [11,12,13,26,31] and the lack of interventions in existing guidelines, it is important to explore which oral health interventions are available for our patient population in existing research. This scoping review will have a broader “scope” with correspondingly less restrictive inclusion criteria. Peters et al. [32] suggests to follow the PCC (Population, Concept and Context) elements. Therefore, the following question based upon the inclusion criteria may be posed: “Which oral health interventions aiming to improve oral health in patients with a mental health disorder are described in existing literature?”

We aim to provide a broad overview of oral health interventions for patients with a mental health disorder including an evaluation of the study quality.

## 2. Materials and Methods

This research aims to provide a broad overview of oral health interventions for patients with mental health disorders and to evaluate the study quality of included studies using Joanna Briggs Institute (JBI) methodology for scoping reviews [32,33] and their checklists [34,35]. A scoping review seeks to provide thorough coverage of literature and is thereby a mechanism for findings for mental health professionals [33]. In contrast to a systematic review, a scoping review adopts more flexibility in study selection, e.g., more flexibility with inclusion and exclusion criteria, and the search terms may be redefined during the process and more criteria can be devised post hoc [32,33].

Although a critical appraisal was not mandatory [36], Brien et al. [37] discussed the lack of quality assessment, and thereby the creation of difficulties in interpretation and conclusion, and a lack of quality also limits the uptake of findings into policy and practice [36]. Therefore, to prevent conclusions based on potential bias, a critical appraisal is legitimised. This scoping review was conducted according to the Preferred Reporting Items for Systematic reviews and Meta-Analyses extension for Scoping Reviews (PRISMA-ScR) Statement [38,39,40].

### 2.1. Stage 1. To Identify the Research Question

The research question for this scoping review was: “Which oral health interventions aiming to improve oral health in patients with a mental health disorder are described in existing literature?”

In this review, ‘oral health’ and ‘mental health disorder’ are defined as follows. In regard to oral health, the definition of Glick et al. [41] will be used. ‘Oral health is multi-faceted and includes the ability to speak, smile, smell, taste, touch, chew, swallow and convey a range of emotions through facial expressions with confidence and without pain, discomfort and disease of the craniofacial complex. Oral health is a fundamental component of health and physical and mental wellbeing. It exists along a continuum influenced by the values and attitudes of individuals and communities; reflects the physiological, social and psychological attributes that are essential to the quality of life; is influenced by the individual’s changing experiences, perceptions, expectations, and ability to adapt to circumstances.’ [41] (p. 229).

Mental health disorder is defined as follows: ‘A mental health disorder is a syndrome characterised by clinically significant disturbance in an individual’s cognition, emotion regulation, or behaviour that reflects a dysfunction in the psychological, biological, or developmental processes underlying mental functioning. Mental health disorders are usually associated with significant distress in social, occupational, or other important activities. An expectable or culturally approved response to a common stressor or loss, such as the death of a loved one, is not a mental disorder. Socially deviant behaviour (e.g., political, religious, or sexual) and conflicts that are primarily between the individual and society are not mental disorders unless the deviance or conflict results from a dysfunction in the individual, as described above’ [42] (p. 20).

### 2.2. Stage 2. Identifying Relevant Studies

The whole point of scoping the field was to be as comprehensive as possible [33]. Therefore, a search strategy was developed in collaboration with a Medical Information Officer (TI) from the University Medical Centre of Groningen (The Netherlands). The search strategy was conducted from the research question and the definitions (Appendix A). Electronic databases MEDLINE, CINAHL, PsycINFO and reference lists were searched from their inception until December 2020. RefWorks Version 2 was used in the study selection process.

For this study, we included peer-reviewed full-text studies published in the English language. Randomised controlled trials (RCT’s), non-randomised intervention studies, observational studies (cohort, case–control and cross-sectional studies), and qualitative studies about oral health interventions in patients with a mental health disorder were included. Systematic reviews and meta-analyses are included, and cross-referencing was applied to search for other relevant articles. Grey literature and guidelines were excluded, because they are composed for knowledge artefacts and were not peer-reviewed [43]. Exclusion criteria were: (i) no focus on an oral health intervention; (ii) an exclusive focus on exploring the severity of oral health problems; (iii) absence of explicit reference to mental health disorders; (iv) primary focus on dementia or mental health retardation; and (v) interventions focusing on the frequency of appointments with the dentist. 

The search identified 1313 potential papers after removing duplicates (Figure 1). Two researchers (S.K. and A.K.-P) screened the abstracts on eligibility based on title and abstract. 

### 2.3. Stage 3. Study Selection

The full texts of the remaining 16 articles were screened (S.K. and A.K.-P.), and 5 articles were excluded as the studies were conference abstracts, with no relevant outcome and protocol of trial (Figure 1). To evaluate the methodological quality, all studies were critically appraised using the checklists developed by the Joanna Brigg’s Institute (JBI). RCTs were assessed with the JBI-tool developed for RCTs consisting of 13 items [34]. Non-randomised intervention studies and pre-test–post-test studies were assessed with the JBI-tool for Quasi-Experimental Studies (9 items) [34]. Cohort studies were appraised with the 11-item JBI-tool developed for cohort studies [35]. Two researchers (S.K. and A.K.-P.) critically assessed the methodological quality independently. Disagreements were discussed. Cohen’s Kappa statistics were calculated to test inter-rater reliability [44]. In case of any disagreement, the aim was to reach consensus with the help of a third researcher (N.B.) during a review meeting. Cross-referencing was applied to search for other relevant articles.

### 2.4. Stage 4. Charting the Data

Following the stages of Arksey and O’Malley [33], the next step was charting the data. In a systematic review, this process is called ‘data extraction’ and was done by two researchers (S.K. and A.K.-P.). In this study, charting the data means recording information relating to the author(s); year of publication; study location; study populations; intervention type and control group (if available); duration of the intervention; aims of the study; methodology; measurement instrument; and important results.

### 2.5. Stage 5. Collating, Summarizing and Reporting the Results

The literature was thematically analysed and from this, groups of mental health disorders, for which the interventions were developed, were distinguished, e.g., SMI (not further specified), psychotic disorders, mood disorders, anxiety disorders, autism spectrum disorder, eating disorders, or substance abuse disorders [42]. The data was abstracted on article characteristics (author, total N, type of study, population, age, gender, type of oral health interventions, outcome, Measurement instrument, assessment time). Next, data on study design and results (intervention/control group, intervention, comparator, results, effect) was summarised.

### 2.6. Ethics and Dissemination

This study did not require ethical approval, as data was collected from existing published peer-reviewed literature and grey literature. The protocol for this review was registered in PROSPERO (ID: CRD42018114415).

## 3. Results

### 3.1. Literature Search

The search yielded a total of 2081 publications as shown in the PRISMA Flow Diagram (Figure 1) [38,39,40]. After removal of duplicates and review of titles and abstracts, 1313 publications were screened for title and abstract. Of these publications, 102 publications were discussed with a third reviewer (N.B.). Finally, 16 full-text publications were assessed for eligibility. Of these, 11 publications were eligible for inclusion and synthesis. Inter-rater reliability for the title and abstract showed a 99.32% agreement with a k = 0.80, demonstrating a high agreement between both raters.

### 3.2. Article Information

With regard to the study design, four RCTs [45,46,47,48], six quasi-experimental studies [28,49,50,51,52,53] and one cohort study [54] were included. Four studies were conducted in Europe, three studies in North America and four studies in Asia. No qualitative studies could be included. The literature was analysed thematically and distinguished groups of mental health disorders for which the interventions were developed: severe mental illness (SMI) or not further specified, psychotic disorders, personality disorders, mood disorders, anxiety disorders, autism spectrum disorder, eating disorders, or substance abuse disorders (Table 1). Data was abstracted on article characteristics (author, total *n*, type of study, population, age, gender, type of oral health interventions, outcome, measurement instrument, assessment time), as included in Table 2. Data on research design and results (intervention/control group, intervention, comparator, results and effect) are summarised in Table 3.

### 3.3. Methodological Quality

All studies (*n* = 11) were included in the methodological quality appraisal. Table 4 identifies the quality appraisal according to the JBI criteria [34,35]. Besides double blindness, three RCTs met all the other JBI criteria [46,47,48]. Additionally, one RCT could also not complete its follow-up assessment [45]. Five quasi-experimental studies met all criteria, except for the control group in the study design [28,50,51,52,53]. One quasi-experimental study met all criteria [49]. The only longitudinal study [54] met four out of eleven criteria. There were no strategies described for dealing with confounding factors and no statistical methods (e.g., regression) were employed to deal with confounding factors [35]. In their study, there were no confounding strategies, reliability and validity of used measurements were not clearly described.

### 3.4. Syntheses: Narrative Summary of Themes

All studies (*n* = 11) focused on patients of which three studies focused on both patients and professionals (Table 1). These studies were focussed on outpatients (*n* = 7) or inpatients (*n* = 4). The literature was analysed thematically, according to the different intervention types, to answer the question regarding what kind of oral health interventions were addressed to improve oral health in adults with a mental health disorder. Four types of oral health interventions emerged from our analysis: (I) educational interventions; (II) physical interventions; (III) interventions combining behavioural and educational elements and IV) interventions combining educational and physical elements.

#### 3.4.1. Educational Interventions

Three quasi-experimental studies [51,52,53] and one cohort study [54] reported on educational interventions in patients with SMI or a mental disorder.

In the study of Barbadoro et al. [51] firstly, inpatients with substance abuse disorders received a questionnaire on socio-economic data and epidemiological data to assess oral hygiene behaviour and other oral health risk factors. After fulfilling the questionnaire, respondents received a complete oral examination according to the WHO criteria (WHO, 1987). A comprehensive oral mucosal examination was performed for evaluating the presence of precancerous lesions. Respondents received a brochure on oral health. A report of the clinical findings was presented to all respondents to promote knowledge about their own oral health status. Results show a significant improvement in exact answers between pre-test/post-test questionnaires (*p* < 0.001), and especially in questions concerning the goals of oral hygiene [51]. One year after the intervention, respondents showed improvement in knowledge and attitude towards oral cancer prevention and toothbrushing had become a daily routine after every meal (in 67.1%). Moreover, 65.9% of respondents had received a dental examination in the previous year. Female gender and more than 10 years of smoking addiction were associated with an improvement of toothbrushing. Respondents with more than 10 years of alcohol addiction were less likely to change [51].

The study of Khokhar et al. [52] focused on outpatients with SMI (not further specified) and professionals. The intervention included the provision of toothbrushes, toothpaste, and mouthwash to respondents without a toothbrush in order to encourage better oral health. Staff was educated on the importance of dental care planning and the inclusion in individual care planning. Access to local dental services was improved. Additionally, respondents received an educational session on dental health. Results of this intervention shows the improvement in access to toothbrushes and the increase of knowledge. The practice of brushing teeth twice a day increased from 29% to 38%. There was no change in number of respondents who had their dentures checked [52].

The study of Silverstein et al. [53] focused on the impact of oral health education in inpatients with eating disorders (anorexia nervosa and bulimia nervosa) to change self-image and oral health practices. Respondents received a pre-test/post-test questionnaire to assess demographics, oral health knowledge, and self-image. The educational programme consisted of three sessions, each with one topic: (i) general oral health education; (ii) aesthetics, effects of eating disorders, pain; and (iii) nutrition for oral health. Every session took 30 min. During the last 10 min of the session, respondents were able to ask questions about their oral health or the sessions’ topic. The results show that patients who reported going to the dentist regularly were significantly more likely to respond that their teeth had a positive effect on their self-image, how they look to others, their general health, and general happiness, compared to respondents who reported going occasionally, or only when they have a problem. Knowledge about general oral health and the impact of an eating disorder on oral health was improved after the oral health sessions.

The longitudinal study of Yoshii et al. [54] focused on outpatients with SMI (psychotic and mood disorders). A pre-/post-programme questionnaire was conducted with demographic characteristics and self-care related to oral hygiene. Additionally, the post-programme questionnaire included items about understanding the educational booklet. The educational programme consisted of five units: cause of tooth loss, dental caries, dental cleaning, periodontal disease and routine dental check-ups. A slide show of 37 slides was presented by a researcher and the handouts were provided. Yoshii et al. [54] used photographic images as much as possible in their educational materials instead of written explanations to make the booklet as impactful as possible. Results showed that the educational programme showed an improvement in the use of fluoride toothpaste, and the daily use of interdental brushes or floss was increased. There were no changes in the frequency of visits to the dentist in the period over 6 months after the intervention. More than 55% of the respondents still went to the dentist when there is a worrying problem or when there are multiple symptoms [54].

#### 3.4.2. Physical Interventions

Mori et al. [50] reported on professional mechanical tooth cleaning (PMTC) in outpatients with an autism spectrum disorder, using an oral examination (consisting of photograph, snap impression, dental radiograph, rough scaling) and the caries activity test, debris index, probing depth, and bleeding on probing to measure the effectivity of interventions. Patients were treated in a dental hospital. The effects of self-care did not significantly change throughout the period of PMTC. The mean probing depth was less than 14 weeks after completion of PMTC, although not statistically significant. The mean number of bleeding sites and debris accumulation was significantly decreased [50].

#### 3.4.3. Interventions Combining Behavioural and Educational Elements

Four RCTs combined behavioural and educational elements in an intervention for patients with a psychotic disorder (*n* = 3) [45,46,47] in patients with a mood disorder (*n* = 2) [46,47], and in patients with SMI (*n* = 1) [48]. In one of these RCTs, educational interventions focused on staff [45]. One quasi-experimental study reported behavioural and educational elements in an intervention in patients with SMI [49].

In the study of Adams et al. [45], Early Intervention Psychosis (EIP) teams received dental awareness training (and an information sheet) during a multidisciplinary team meeting (approximately 30 min). This training for teams included information about the trial and checklists for patients. Patients received a dental checklist in order to improve oral health behaviour. However, due to missing data in follow-up (e.g., high turnover of staff members), evidence could not be studied [45].

In the study of Almomani et al. [46], outpatients with psychotic or mood disorder received dental education, oral hygiene instructions, and reminders (a reminder system and a once-a-week phone call) from a dental hygienist to provide positive feedback and to underline the study instructions. The effects of these interventions were measured with the plaque index score (pre- and post-intervention). Results show that oral health in the intervention group improved significantly regarding plaque accumulation and knowledge level relative to the members of the control group (F = 5.32, *p* = 0.026, η^2^ = 0.1), who only received a mechanical toothbrush [46].

In the study of Almomani et al. [47], outpatients with SMI (schizophrenia, bipolar disorder and depression) received brief motivational interviewing (MI) sessions (15–20 min, frequency is unknown) prior to an educational session (with focus on information about the effects of SMI on oral health, exploring advantages of good oral hygiene and disadvantages of bad oral hygiene, motivation, confidence, and personal values related to oral health). Patients (intervention and control group) received pamphlets with information from the educational session and an instruction on how to use a mechanical toothbrush, a reminder system, and once-a-week telephone calls [47]. In the MI group, oral health knowledge in the intervention group improved significantly 4–8 weeks after baseline. The MI group showed significantly less plaque 8 weeks after baseline compared to the education group. Additionally, on the plaque index, there was a large interaction effect (η^2^ = 0.8). One of the limitations in this study is the lack of follow-up over a two-month time frame. Additionally, it is not known if these effects would be maintained over an extended period. 

One study reported an RCT with an oral health promotion programme with group and individual components in outpatients with SMI [48]. In this 12-week experimental study, the intervention group received group oral health education in five sessions with an interval of 2 weeks. The intervention group received pictures of the Bass toothbrushing technique procedure that were posted on the mirror in the bathroom of participants. Songs with a message of the benefits of toothbrushing were broadcasted five times each day (upon waking up, after each meal, and before going to sleep) [48]. The individual interventions included instructions from a trained nurse. Tokens were used after the nurses had checked the accomplishment of successful toothbrushing. The control group received usual nursing care. After 12 weeks, the mean dental plaque index significantly improved in the intervention group (*p* < 0.001). Oral health knowledge, oral health attitude, and oral health behaviour were statistically improved in the intervention group after 12 weeks (*p* < 0.001). Consumption of sugary beverage and dentist-visiting behaviour did not show a significant change [48].

One quasi-experimental study reported an intervention combining behavioural and educational elements in patients with SMI [49]. In this study, with random assignment at the level of treatment centre, participants were assigned to four study groups. Group A received oral hygiene education and a battery-operated toothbrush. Group B received only the same battery-operated toothbrush as Group A. Group C received the oral hygiene education as well as a manual toothbrush. Group D received only the same manual toothbrush as Group C. Participants in Groups A and C received oral hygiene instructions [54]. Participants were observed while performing oral hygiene with their toothbrushes in the dental chair of their dentist. Dental plaque index and individual modifications were made for participants in Groups A and C when needed. All participants were provided with an evaluation and instruction per visit. They received and were instructed to use Crest Cavity Protection toothpaste for the duration of the study. Participants received a calendar with the four study weeks and stickers were provided to each participant. Participants were asked to perform oral care twice daily (morning and before bedtime) and affix a sticker to the calendar for that particular day. Smoking status (current smoker and never smoked) was assessed [49]. A statistically significant effect is found on the type of toothbrush participants used (*p* < 0.05). Interaction of home care instructions and type of toothbrush were not found. This study showed a significant effect on gingival index associated with the mechanical toothbrush (*p* < 0.05). No statistically significant changes were found in plaque index based on type of toothbrush. The provision of oral home care instructions showed no significant difference in the mean change in plaque and gingival index. There was no correlation between the negative symptoms and the post-test mean plaque index and the post-test gingival index. Frequency of brushing and the mean change of plaque index and gingival index were not correlated. There was no significant impact of smoking on the mean change in plaque index and gingival index.

#### 3.4.4. Interventions Combining Educational and Physical Elements

One quasi-experimental study reported about an intervention combining educational and physical elements in professionals (nurses) and SMI patients (psychotic, personality, mood, anxiety, and autism spectrum disorders) [28].

Nurses received education to improve knowledge and awareness [28]. Nurses were educated in oral health and instructed to help patients, using an instruction card with oral hygiene tips and oral hygienists demonstrated cleaning methods and tools (e.g., interdental cleaning aids). Outcome measurements in professionals were oral health knowledge of nurses (a 20-item-list on proper oral care, oral diseases, and oral health-related factors, with an internal consistency of α .62, which was moderate). Patients with SMI received a patient treatment plan from the oral hygienist after a baseline oral examination. They received a soft toothbrush, fluoridated toothpaste, specific cleaning instructions from the oral hygienist (e.g., brush at least 2 min, brush systematically), and practiced the instructions under the supervision of the oral hygienist [28]. Outpatients were asked questions about medical and general health, dental and oral health, and condition. After the questionnaire, the oral hygienist carried out pre-test and post-test measurements of the dental plaque and gingival bleeding indices. The results showed a significant effect on plaque index/6 (r = 0.82) and index/2 (r = 0.77) and bleeding index/6 (r = 0.50).

The educational intervention had elicited a statistically significant change in mental health nurses’ knowledge (*p* < 0.001). One of the limitations reported in this study was the low active commitment of nurses. Some nurses actively applied motivational interventions and others were scarcely involved in actively promoting oral health among patients. Nurse’s activity was not systematically monitored and registered.

## 4. Discussion

This scoping review was conducted to provide a broad overview of oral health interventions in mental health and to evaluate the study quality. The review demonstrates that little has been developed in order to improve oral care for people diagnosed with mental health disorders, despite the fact that this is an important topic that influences all aspects of quality of life. Eleven studies were included that reported interventions focusing on behavioural, educational or physical interventions, or combinations of these aspects for patients and/or professionals. All studies (*n* = 11) had an evaluation period ≤ 12 months. Nine studies showed an effect on the short term (≤12 months) with regard to oral health knowledge, oral health behaviour, or physical oral health outcomes (e.g., plaque index). Two studies showed no effects on any outcome. In general, the methodological quality was moderate to sufficient. Overall, this review demonstrates that educational, behavioural, and physical interventions or combinations of these elements have a positive effect on oral health knowledge (N = 5 out of 11 studies), the frequency of brushing (access to toothbrush, toothpaste) (N = 3), plaque index (N = 4) and gingival bleeding on probing depth (N = 4). No significant differences were measured on dental visits and the consumption of sugary drinks.

The positive effects were found in patients diagnosed with SMI, psychotic disorders, personality disorders, mood disorders, anxiety disorders, autism spectrum disorders, eating disorders, or substance abuse disorders—with different outcomes. Of all studies, seven studies were focussed on outpatients, four studies were focussed on inpatients. Looking for interventions, it is important to look carefully whether interventions are developed for inpatients or outpatients, because the findings may not be generalizable to all patients with a mental health disorder.

The results of this study makes it difficult to draw any firm conclusions on which intervention works for whom and which elements should be part of effective interventions. Based on the results of this study, due to the heterogeneity in both interventions, diagnostic groups and outcomes, one golden standard oral health intervention cannot be advised yet, although the methodological quality of studies seems sufficient.

Research in patients with a severe mental illness (SMI) demonstrated that 58% of the patients had low oral health-related quality of life (OHRQoL) [55]. This supports the importance of oral health interventions in patients diagnosed with a mental health disorder.

### 4.1. Reflection on Types of Interventions

Firstly, it is remarkable that none of the studies in our scoping review focused on preventive care. Secondly, knowledge on oral health in patients and nurses is quite well-studied, however research into behavioural change in patients as outcome, as well as in mental health professionals, is also desirable. Knowledge and awareness are a first step; however, behaviour change in on oral health is a complex process that requires another approach.

Reminder strategies combined with oral health education showed to have a significant effect on behaviour of patients with a mental health disorder (schizophrenia, depression, bipolar disorder) [46]. Reminder systems, such as post-it, are easy to implement. Alqahtani et al. [56] show that reminder strategies enable a system to remind the user to perform the target behaviour. Reminders are often implemented to remind users to perform activity in mental health disorders [56]. There are no studies examining the effects of reminder strategies focusing on oral health in mental health apps. Therefore, further research on reminder strategies improving oral health is needed.

Overall, educational interventions significantly improve knowledge of mental health professionals on general health and the importance of oral healthcare. The combination of educational interventions and behavioural interventions are only studied in patients with psychotic and bipolar disorder. These studies demonstrate that the combination of these elements is effective on oral hygiene (as measured with the plaque index) and oral health knowledge [46,47]. Oral health interventions using physical elements were only studied in patients with autism spectrum disorders [50]. These patients benefitted by a decrease of bleeding sites; however, 14 weeks after this intervention, there was no longer a post-treatment effect. This is in line with the study of Kay and Locker [57], who discussed the short-term effects of oral health interventions, although this systematic review only shows the evidence on dental health education until 1996. Additionally, it shows evidence in the general population and does not have a focus on dental health education in patients with a mental health disorder. Is does, however, display the importance of continuity in treatment, as well as long-term monitoring. Future research might take this into account.

Included studies were focused on different interventions based on literature. Additionally, mental health nurses and patients with a mental health disorder should play an important role in the development of interventions so that an appropriate approach can be developed in co-creation with the end-users.

### 4.2. Quality of Included Studies

Interventions combining behavioural and educational elements appear to be effective in patients with SMI (diagnosis not further specified), psychotic disorder, and mood disorder. Of these studies, the methodological quality was good (*n* = 3) [46,47,48]. In one RCT, the methodological quality was moderate due to insufficient follow-up data [45]. In three RCTs, there was no blinding of participants and outcome assessors. Blinding is a measure in RCTs to reduce detection and performance bias and is an important measure in RCTs. There is evidence that lack of blinding leads to overestimated treatment effects. If participants are not blinded, knowledge of group assignment may affect participants behaviour in an RCT [58]. This means that the treatment effects in the included RCTs [45,46,47] can be overestimated. Blinding outcome assessors can be used in order to minimise distortion in the results of the study [34]. Detection bias can arise if the knowledge of patient assignment influences the assessment of outcome measurements. This detection bias can be avoided by the blinding of those assessing outcomes in an RCT [58,59]. For included RCTs, it is not known if knowledge of a patient’s assignment had influenced outcome measurements.

Quasi-experimental studies show the effectiveness in interventions combining educational and physical elements in patients with a psychotic disorder, personality disorder, anxiety disorder, mood disorder and autism spectrum disorder [28]. Educational interventions appear to be effective in patients with SMI (diagnosis not further specified), eating disorder and substance abuse disorder [49,51,52,53]. Physical interventions appear to be effective only in patients with autism spectrum disorder [50]. The methodological quality of the quasi-experimental studies was sufficient. However, the pre-test–post-test design of the studies did not aim to compare an intervention group with a control group. The addition of control groups and sensitivity analyses can support the hypothesis that the intervention is causally associated with the outcome [60]. One 2 × 2 quasi-experimental study met all the requirements of the JBI checklist; however, this study of Singhal et al. [49] lacked the determination of the effect of the calendar. It is not known if there was a Hawthorne effect and if the calendar was a confounder for other independent variables [49]. Furthermore, SMI has to be specified because every patient group has its own needs in managing oral health problems. Additionally, physical interventions should also be examined in other mental health patient groups beside ASD (Autism Spectrum Disorder).

The quality of the cohort study (educational intervention [54]) was insufficient as confounders were not clearly identified and no strategies dealing with confounding factors were described [35]. In the literature, confounding has been described as a confusion of effects [58]. To draw appropriate conclusions about the effect of the educational intervention on an outcome, the causal effects should be separated from that of the other factors that affect the outcome (e.g., age) [61]. Strategies (e.g., matching, randomization, stratification) were not used in this cohort study [62]. Due to the lack of controlling for confounding factors in included cohort study [54], it is not clear whether the conclusions were drawn appropriately or that there were other factors that affect the outcome measurements. 

### 4.3. Reflection on the Effect of Interventions

Overall, dental health education or lectures, dental care instructions, brief motivational interviewing, and a reminder system or a treatment plan showed a significant and positive effect on oral health knowledge, Q.H. plaque index, or TRSQ. The use of one of these interventions, combined with a mechanical toothbrush, can improve the oral hygiene of people with mental health disorders. Providing patients with a toothbrush, toothpaste, and mouthwash was helpful to increase access to toothbrushes and brushing twice a day and had a significant effect on plaque index. PMTC showed a significant decrease in probing depth and total number of bleeding sites; however, no significant change on caries activity test and debris accumulation. There was no significant impact of smoking on the mean change in plaque index and gingival index. These effects were tested in patients diagnosed with SMI (psychotic disorders, personality disorders, mood disorders, anxiety disorders, autism spectrum disorders, eating disorders, and substance abuse disorders). There are constantly new insights regarding oral health. A recent study for example showed the effectiveness of a mechanical and ultrasonic toothbrush on oral biofilm removal [63]. This highlights the proactive approach for clinical and home management through the use of mechanical or ultrasonic toothbrushes in outpatients and inpatients with a mental health disorder. Furthermore, a recent study on students of Lee et al. [64] showed that ingestion of the oral probiotic Weissella cibaria can help reduce subjective halitosis and improve oral-health-related quality of life. However, this was not tested as intervention in patients with a mental health disorder. Therefore, further research on the use of oral probiotic Weissella cibaria could be interesting.

A significant effect associated with the mechanical toothbrush is found on gingival index [50]. The problem that arises in these studies, is the unknown effects of dental health education and behavioural interventions over a longer period of time. Interventions were measured during a period between four weeks and ≤12 months. Kay and Locker [57] concluded in a systematic review about dental health education among the general population that effects are probably short-lived. However, this study is not very recent and focussed on the general population, and not specifically on people with a mental health disorder. Long-term effects of oral health interventions in patients with a mental health disorder are not known. Thus, future studies should consider measuring the impact of oral health interventions on oral health status as well as knowledge and behaviour changes over a longer period of time, in line with and depending on the needs of patients.

### 4.4. Competences of Mental Health Professionals in Studied Interventions

Professionals who were involved in care for patients in the included studied were nurses. According to De Mey et al. [28] the non-participation of mental health nurses is a concern. At the end of their study, 50% of the mental health nurses took part in the project and the active commitment was even lower. To care for and motivate patients regarding their oral health is imperative, and part of mental health nurses’ tasks in their daily contact with patients with SMI. Recent research shows that patients after a first episode psychosis or SMI are not always able to take care of their oral health, and this should be included in the daily work of nurses [13]. Mental health nurses do not apply motivational and supportive interventions concerning patients’ oral health and mental health nurses were scarcely involved in actively promoting better oral health [28]. In their daily care for patients, it is essential that nurses take oral health care in patients into account. Studies show the importance of training nurses in promoting better oral health, although this study also confirms our concern about the participation of mental health nurses in oral health. These concerns about the involvement of mental health professionals are in line with literature that states that mental health professionals do not routinely practice oral care [65]. The question arises if mental health nurses have sufficient knowledge about oral health care. The attitude of nurses towards personal health shows that nurses prioritise symptoms of mental health illnesses instead of risk factors and consequences. Adams et al. [45] advises further research to understand the barriers for mental health nurses to manage oral health needs of patients.

Adams et al. [45] found a lack of research culture and a high turnover rate within teams and that the initial enthusiasm for the RTC could not be sustained. One of the reasons for failure is the lack of ownership in study and design within service-users and clinicians who designed the study, due to the top-down changes in the team. Therefore, it is important to facilitate one or two nurses (or nurse practitioners) with the task of care-coordinator with special attention for oral health. To date, studies have not taken this into account. 

Further studies regarding mental health nurses should consider mental health nurses’ attitude and barriers towards oral health care.

### 4.5. Study Strengths and Methodological Considerations

This scoping review has some methodological considerations worth noting and provides information for future nursing research regarding oral health care and interventions in patients with a mental health disorder. This study is strengthened by the assessment of the study quality of included studies, which was not necessary because this was a scoping review [66]. The belief was that it was important to prevent the drawing of conclusions based on potential bias, and that a critical appraisal was legitimised. Moreover, this study is strengthened by reducing potential bias through calculating the inter-rater reliability and the involvement of two reviewers in the selection process.

This study has some methodological considerations. Firstly, studies that varied in population (patients diagnosed with different disorders), intervention (different interventions were tested), and quality (some results show missing data in follow-up for several reasons) were included. Therefore, the results may have decreased generalisability. It is not clear whether the study effects in autism or SMI will also be generalisable to all patients with a mental health disorder, since most of them were developed for specific target groups. Secondly, in this study, all results regarding oral health interventions for a mental health disorder are summarised; however, it is unknown whether elements are generic or specific. Thirdly, as is the case in every review, it was possible that negative results regarding oral health interventions in patients with mental health disorders are missed due to publication bias (e.g., exclusion of abstracts for conferences or study protocols). Additionally, due to the exclusion of grey literature, it was possible that we missed interventions that are described, but not published yet. This may have affected the results and overall conclusions of this study. However, to make our scoping review and the critical appraisal more feasible for clinical practice, we decided to include peer-reviewed articles from electronic databases. Further adding to this issue is that many grey literature contain information that is not publicly readable or available. Fourthly, due to the limitation of publications in English, it is possible that we missed peer-reviewed studies on oral health interventions in other languages. However, it is unknown if this affected the results and overall conclusions of this study.

Despite these limitations, the review does provide important understandings of oral health interventions in patients diagnosed with a mental health disorder.

## 5. Conclusions

Prior literature has examined educational, behavioural, and physical interventions in order to improve oral health among patients diagnosed with a mental health disorder. An important conclusion of this review is that despite the importance of paying attention to good oral hygiene, very little oral health interventions are developed for patients with a mental health disorder. There is no golden standard that can be recommended at this moment. To date, mental health professionals, and especially nurses, are the group that support patients in their daily activities (e.g., activities of daily living and lifestyle), so they are the primary target group that can influence oral health in patients with mental health disorders. When caring for patients diagnosed with a mental health disorder, it is essential that mental health professionals consider oral health care as an essential part of their daily tasks and provide necessary nursing support. Mental health nurses should be more aware of oral health and oral health risk and should provide long-term interventions in order to improve oral health. Further research into the current competences of nurses to support and motivate patients with a mental health disorder and to be able to apply oral health interventions is conditional.

To develop and implement more solutions that are suitable in view of future research, it seems important to develop an integrated toolkit with interventions, in which all these components are given a place, as well as interventions for preventive care. Oral health programmes should be provided tailored to the needs of the patient.

## Figures and Tables

**Figure 1 ijerph-18-08113-f001:**
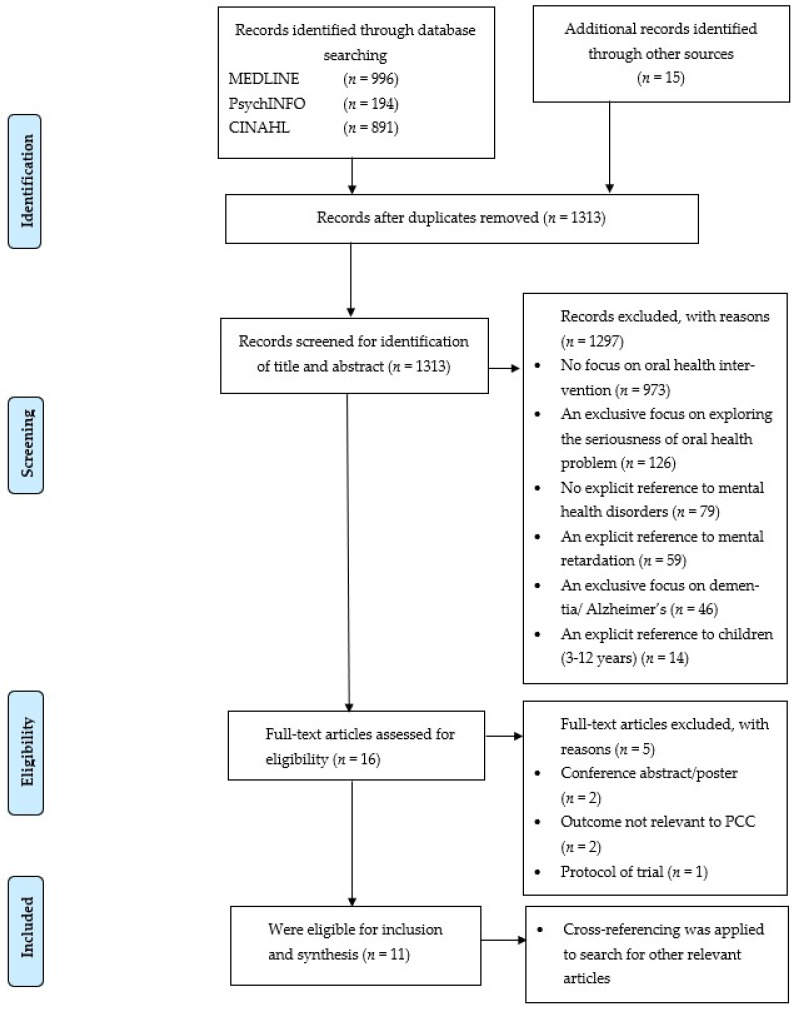
PRISMA flow diagram for the scoping review process [38,39].

**Table 1 ijerph-18-08113-t001:** Overview of oral health interventions in mental health: type of intervention (I-IV), study participants (professionals or patient group) and study design.

Author	I Educational Intervention										II Physical Intervention										III Interventions Combining Behavioral and Educational Elements										IV Interventions Combining Educational and Physical Elements									
	Prof		Patient Group								Prof		Patient Group								Prof		Patient Group								Prof		Patient Group							
			SMI	PSD	PD	MD	AD	ASD	ED	SAD			SMI	PSD	PD	MD	AD	ASD	ED	SAD			SMI	PSD	PD	MD	AD	ASD	ED	SAD			SMI	PSD	PD	MD	AD	ASD	ED	SAD
**Randomised controlled trials**																																								
Adams et al. [45]																					x			x																
Almomani et al. [46]																								x		x														
Almomani et al. [47]																								x		x														
Kuo et al. [48]																							x																	
**Quasi-experi-mental studies**																																								
Barbadoro et al. [51]										x																														
Khokhar et al. [52]			x																																					
De Meij et al. [28]																															x			x	x	x	x	x		
Mori et al. [50]																		x																						
Silverstein et al. [53]									x																															
Singal et al. [49]			x																																					
**Cohort study**																																								
Yoshii et al. [54]			x	x		x																																		

Abbreviations: Prof: professionals. Patient disorder: SMI: Severe Mental Illness not further specified—PSD: Psychotic Disorder—PD: Personality Disorder—MD: Mood Disorder—AD: Anxiety Disorder—ASD: Autism Spectrum Disorder—ED: Eating Disorder—SAD: Substance Abuse Disorder.

**Table 2 ijerph-18-08113-t002:** A summary of the general characteristics, outcomes and measurements of the included studies.

First Author, Year Publication	Total N	Type of Study	Population	Age in Years	Gender % Man	Setting (Recruitment)/Country	Type of Oral Health Interventions	Outcome	Measurement Instrument	Assessment Time
Adams et al. [45]	35 EIP teams and their service users (N = > 1682)	RCT	Suspected psychosis Outpatients	15–56	66%	EIP Teams, Manchester, United Kingdom	Interventions combining behavioural and educational elements	Behaviour towards oral health	Oral health: OIDP checklist Behaviour: general questionnaire: registered with dentist, routine check-up	Baseline, 12 months
								Knowledge: dental awareness training	Owning a toothbrush, cleaning teeth twice a day, urgent dental treatment.	
Almomani et al. [46]	N = 50	RCT	Schizophrenia Bipolar disorder, depression Outpatients	19–61	46%	Communitysupport programme, Kansas, USA	Interventions combining behavioural and educational elements	Behaviour: oral health instruct-tions and reminder system. Knowledge: dental education	Plaque: Quigley-Hein plaque index. Knowledge: Questionnaire oral hygiene	Baseline, 4 weeks
Almomani et al. [47]	N = 60	RCT	Schizophrenia Bipolar disorder, depression Outpatients	22–58	50%	Community support programme, Kansas, USA	Interventions combining behavioural and educational elements	Behaviour in oral health: MI Knowledge on oral health.	Behaviour: TRSQ Plaque: Quigley-Hein plaque index. Knowledge: 15-item oral health knowledge questionnaire.	Baseline, 4 weeks, 8 weeks
Kuo et al. [48]	N = 58	RCT	SMI Inpatients	20–80	100%	Two psychiatric wards of a general hospital, Taiwan	Interventions combining behavioural and educational elements	Plaque accumulation	Plaque: Plaque control record	12 weeks
								Oral health promotion programme: oral health knowledge, attitude and behaviour	Knowledge, behaviour, attitude: 35 item questionnaire	
Barbadoro et al. [51]	N = 76	QES	Alcohol-addiction Inpatients	Not clear	76.3%	Residential rehabilitation clinic, Italy	Educational intervention	Knowledge on oral health and risk factors.	Knowledge: 10-item test assessing knowledge and consciousness	12 months
Khokhar et al. [52]	N = 59	QES	SMI Inpatients	22–76	68%	Heather Close Recovery Unit, Mansfield, UK	Educational intervention	Knowledge on importance of dental care	General questionnaire: Access to toothbrushes, registered at dentist, dentures checked in last 5 years. Knowledge of basic oral hygiene.	12 months
De Mey et al. [28]	N = 27 (Pr) N = 24 (P)	QES	Psychotic, Personality Mood, Anxiety, Autism disorders	22–69	Not known	Mental health organisation, the Netherlands	Interventions combining educational and physical elements	Knowledge nurses: oral care and tools, diseases, alcohol, smoking and drugs	Nurses knowledge: 20-items knowledge of oral hygiene.	Baseline, 5 weeks
			Outpatients					Oral health in patients.	Oral health: Patients: Dental plaque index. Gingival bleeding index	
Mori et al. [50]	N = 10	QES	Autism, mental health retardation, Hydrocephalus Outpatients	21–29	90%	Special care dentistry, Osaka University Dental Hospital, Japan	Physical intervention	Oral health: reduction of bleeding sites on probing	Caries activity test (pH-meter) Debris index, Probing depth, bleeding on probing	Baseline, 2 weeks, 6 weeks, 14 weeks
Silverstein et al. [53]	N = 67	QES	AN-BP and BN Inpatients	13–50		Hospital eating disorder clinic, North Carolina	Educational intervention	Knowledge on oral health and habits, hygiene practices.	Oral health knowledge, oral habits, oral health behaviours and habits since diagnosis, self-perception.	Baseline, and after following the programme.
Singhal et al. [49]	N-87	QES	SMI Outpatients	18–83	Not known	Rural and urban outpatient mental wellness centre, New Jersey USA	Interventions Combining Behavioural and Educational Elements	Oral hygiene education and battery-operated toothbrush or manual toothbrush	Oral health: Quigley-Hein plaque index and gingival index	Baseline, 12 weeks
								Level of negative symptoms related to SMI	Self-evaluation of negative symptoms survey (SNS)	
Yoshii et al. [54]	N = 390	Cohort study	Mental illness, Psychotic and mood disorders Outpatients	20–80	Not known	Psychiatric day-care centres, Japan	Educational intervention	Knowledge: cause of tooth loss, dental caries, dental cleaning periodontal disease and routine dental check-ups.	Knowledge: 20-item selfcare questionnaire	Baseline, 1 week, 1 month, 3 months, 6 months.

Abbreviations: total N—PR: professionals—P: Patients. Type of study—RCT: Randomised Controlled Trial—QES: Quasi-Experimental Study. Population—SMI: Severe Mental Illness—AN-BP: patients with Anorexia-binge eating/purging—BN: Bulimia Nervosa. Measurement—OIDP: Oral impact on daily profile—TRSQ: Treatment Self-regulation Questionnaire—DMFT: Decayed, Missing, and Filled Teeth—EIP: Early Intervention Psychosis.

**Table 3 ijerph-18-08113-t003:** Summary of results of oral health interventions in mental health.

First Author, Year Publication	Intervention Group (N)	Control Group (N)	Intervention	Comparator	Results	Effect +/−
Adams et al. [45]	18 EIP teams	17 EIP teams	Dental awareness training. Dental checklist. Oral hygiene information sheet with oral hygiene tips and information on how to find a dentist.	Standard care. One year after intervention: Dental awareness training, checklist and oral hygiene information sheet.	No significant differences were found in: registered with dentist (*p* = 0.44), routine check-up (*p* = 0.18), owning a toothbrush (*p* = 0.99), cleaning teeth twice a day (*p* = 0.68), urgent dental treatment (*p* = 0.11), OIDP checklist: no prospective data collected.	−
Almomani et al. [46]	N = 20	N = 22	Dental education. Oral hygiene instructions. Mechanical toothbrush. Reminder system.	Mechanical toothbrushes.	Q.H. plaque index: The improvement in the intervention group was significantly higher than the control group (*p* = 0.026). Of them, 95% reported that reminders and oral health promotion were helpful	+
Almomani et al. [47]	N = 30	N = 30	Brief MI sessions on motivation and confidence, personal values. Educational sessions: exploring advantages and disadvantages oral hygiene, effects of SMI on oral health. Two pamphlets summarizing the info from education and instruction in using a mechanical toothbrush. Weekly phone calls (for 4 weeks).	Educational sessions: exploring advantages and disadvantages oral hygiene, effects SMI on oral health. Two pamphlets summarising the info from education and instruction in using a mechanical toothbrush. Weekly phone calls (for 4 weeks).	Q.H. plaque index: scores from the inter-vention group were improved from base-line to 4 wks (*p* < 0.01) and from 4–8 wks (*p* < 0.01) and had significantly less plaque than control group after 8 weeks (*p* < 0.01). TRSQ: Oral health knowledge improved in both groups from baseline to 4 wks (*p* < 0.01). From 4–8 wks, the improvement in the inter-vention group was significantly higher (*p* < 0.01).	+ +
Kuo et al. [48]	N = 27	N = 31	Oral health programme: group education in 5 sessions: structure of oral cavity and teeth; importance of oral health; pathogenesis of caries and periodontal diseases; Bass toothbrushing method; and oral hygiene. Pictures of toothbrushing methods were posted on the mirror in each bathroom. Individual instructions in Bass toothbrushing method and one-on-one training in Bass toothbrushing technique were given. Participants were checked for correctness. Individual behavioural modification: participants received tokens for successful brushing.	Nursing care as usual	After 12 weeks, the mean dental plaque index significantly improved, compared to that of the control group (*p* < 0.001). Oral health knowledge, oral health attitude and oral health behaviour were statistically significant improved after 12 weeks, compared to those in the control group (*p* < 0.001). No significant differences between intervention and control group on consumption of sugary beverage and dentist-visiting behaviour after the intervention.	+ + −
Barbadoro et al. [51]	N = 76	No control group	After oral examination with DMFT: participants received a report with clinical findings. Lecture about alcohol, tobacco smoke in oral health pathology, oral cancer prevention strategies (de, secondary and tertiary). Brochure on oral health.	No comparator.	10-item test assessing knowledge and consciousness: there was an improvement of 25% in exact answers between the pre-test/post-test (*p* < 0.001). Participants showed a significant improvement in toothbrushing after every meal (*p* < 0.001). Female, age >49 year, years of alcohol addiction had more risk of high DMFT (4.33/1.85/6.13).	+
Khokhar et al. [52]	N = 59	No control group	Staff: education on importance of dental care and inclusion in care planning. Patients: provided with toothbrushes, toothpaste and mouthwash. Informa-tion and advice on basic dental hygiene by visual aids, posters and demonstration models. Registration status with dental practices. List of local dental practitioners was provided.	No comparator.	Access to toothbrushes increased from 68% to 86%. Knowledge of basic oral hygiene was improved from 55% to 61%. Brushing twice daily increased from 29% to 38%. There was a small increase from 34% to 39% of patients registered at dentist. There was no change in patients who had their dentures checked within the last 5 years.	+
De Mey et al. [28]	N = 27 (Pr) N = 24 (P)	No control group	Nurses: PowerPoint presentation about oral care, available tools, oral diseases (gingivitis, periodontal disease and caries), and oral health related to smoking, alcohol and drugs. Cleaning methods and tools were demonstrated by an O.H. Patients: an O.H. set up a treatment plan after oral examination. Toothbrush and fluoridated toothpaste. Concrete instructions on brushing and cleaning. Images were used of the toothbrush in different positions. Instruction card.	No comparator	Nurses: knowledge was significantly improved (*p* < 0.001). Patients: Dental plaque index was significantly improved on plaque index/6 (*p ≤* 0.001), and plaque index/2 (*p* < 0.001). Concerning gingival bleeding index there was a significant change (*p* < 0.05). No significant changes were observed in bleeding index/2.	+
Mori et al. [50]	N = 10	No control group	Six sextant evaluation for PMTC. Six teeth were stained with Red Coat and DI. PMTC procedure was strictly according Axelsson’s method.	No comparator.	Probing depth was significantly decreased at PMTC VI (*p* < 0.05). A reduction of the total number of bleeding sites on probing was significantly different (*p* < 0.05).	+
Silverstein et al. [53]	N = 67	No control group	Educational programme (Smiles Matter). Weekly presentations: general oral health education, aesthetics, effects of eating disorders, oral pain, nutrition for oral health.	No comparator.	Patients who reported regular visits to the dentist were significantly more likely to respond that teeth had a positive effect on how they looked to themselves (*p* = 0.03), looking to others (*p* = 0.03), kissing (*p* = 0.04), their general health (*p* = 0.01), romantic relationships (*p* = 0.04) and general happiness (*p* < 0.001).	+ +
Singhal et al. [49]	N = 41	N = 47	Group A: oral hygiene education and a battery-operated Arm and Hammer Truly Radiant Spin Brush. Group C: oral hygiene education as well as the Sun Star Gum ultrasoft manual toothbrush. C. Participants in group A and C received oral hygiene instructions and included video demonstrations. Observation were made while performing oral hygiene with their toothbrushes sitting in the dental chair. Dental plaque index and individual modifications were made.	Group B only received the same battery-operated toothbrush as group A. Group D received the same manual toothbrush as Group B. Next, participants received and were instructed to use Crest Cavity Protection toothpaste. A calendar and stickers were provided.	A statistically significant effect is found on the type of toothbrush participants used (*p* < 0.05). Interaction of home care instructions and type of toothbrush were not found. A significant effect is found on gingival index associated with the mechanical toothbrush (*p* < 0.05). No statistically significant changes were found in plaque index based on type of toothbrush. The mean change in plaque and gingival index were not significantly different based on the provision of oral home care instructions.	+ − + − −
			Evaluation and instruction per visit. Participants received and were instructed to use Crest Cavity Protection toothpaste. A calendar and stickers were provided. Participants were asked to perform oral care twice daily (morning and before bedtime) at home for four weeks and affix a sticker to the calendar for that particular day.	Participants were asked to perform oral care twice daily (morning and before bedtime) at home for four weeks and affix a sticker to the calendar for that particular day.	There was no correlation between negative symptoms and the post-test mean plaque index and the post-test gingival index. Frequency of brushing and the mean change of plaque index and gingival index were not correlated. There was no significant impact of smoking on the mean change in plaque index and gingival index.	− − −
Yoshii et al. [54]	N = 390	No control group	Educational programme: (1) cause of tooth loss, (2) dental caries, (3) dental cleaning, (4) periodontal disease, (5) routine dental check-ups. This was in a 30 min-slideshow of 37 slides. Photos of patients’ mouths were used.	No comparator.	The educational programme showed a significant improvement in the use of fluoride toothpaste at 6 months after the intervention (*p* = 0.001). The daily use of interdental brushes or floss was significantly improved 6 months after the intervention (*p* = 0.025). There was no change in frequency of visits to the dentist.	+ −

Abbreviations: intervention group—EIP: Early Intervention Psychosis—PR: professionals—P: Patients. Intervention—MI: Motivational Interviewing—SMI: Severe mental illness—DMFT: Decayed, Missing, and Filled Teeth—O.H.: Oral Hygienist—PMTC: Professional mechanical tooth-cleaning—DI: debris index. Results—OIDP: Oral impact on daily profile—Q.H. plaque index: Quigley–Hein plaque index. Effect: − = No significant effect, + = significant effect.

**Table 4 ijerph-18-08113-t004:** Critical appraisal of selected studies on oral health interventions in mental health.

Randomised Controlled Trials ^1^												
	Randomi-sation for assignment to treatment group	Allocation conceal-ment	Similar treatment groups at baseline	Participants blind totreatment assignment	Delivering treatment blind to treatment assignment	Outcome assessors blind totreatment assignment	Treatment groups treated identically	Follow-up complete or differences adequately analysed	Analysed in groups to which they were randomised	Outcomes measured in same way	Outcome measure-ments reliable	Appro-priatestatistical analyses used	Appro-priate design
Adams et al. [45]				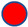	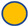	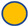		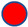					
Almomani et al. [46]		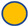		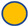	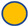	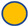							
Almomani et al. [47]		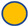		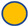	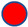	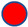							
Kuo et al. [48]				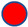	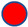								
**Quasi-experimental studies ^2^**												
	Are the‘cause’ and the ‘effects’ clear	Participantsincluded in comparisons similar	Receiving simi-lar treatment, other than intervention	Was there a controlgroup?	Multiple measurementsof the out-come pre- and post	Follow-up complete or differences described	Outcomesmeasured in same way	Outcome measure-ments reliable	Appropriate statistical analyses used				
Barbadoro et al. [51]				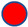									
Khokhar et al. [52]				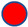									
De Mey et al. [28]				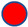									
Mori et al.[50]				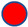									
Silverstein et al. [53]				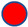									
Singhal et al. [49]													
**Cohort study ^3^**												
	Groups simi-lar and recruited from the same population	Exposures measured similarly toassign people	Exposures measured valid and reliable	Confoun-ding factors identified	Strategiesto deal with confounders stated	Groups freeof the out- come at the start of the study	Outcomes measured in a valid and reliableway	Was follow-up time reported and sufficient	Follow-up complete,reasons to loss explored	Strategies to address incomplete, follow-up utilised	Appropriate statistical analyses used		
Yoshii et al.[54]	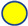	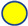	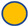	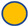	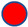		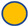			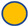			

^1^ As measured with the JBI critical appraisal tool for randomised controlled trials [34]; ^2^ as measured with the JBI critical appraisal tool for quasi-experimental studies [34]; ^3^ as measured with the JBI critical appraisal tool for cohort studies [35]. Note: 

 = yes, 
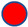
 = no, 
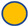
 = unclear, 
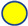
 = not applicable.

## Data Availability

Data is available from corresponding author upon request.

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
