# Peer review of "Oral Health Interventions in Patients with a Mental Health Disorder: A Scoping Review with Critical Appraisal of the Literature"

_ijerph, 2021, doi:10.3390/ijerph18158113_

Round 1
Reviewer 1 Report
Congratulations for the hard work. I suggest some minor revision
- Update the PROSPERO record, as important methodological changes were made to the initial proposal
- Include the acronym PCC (Population, Concept, and Context) after your study question
- In the Flow diagram, check the number of Records screened for identification of title and abstract (is it the same number found after duplicates removed?)
- Conduct a more in-depth discussion of the quality analysis, as the problems were raised, but they were not always sufficiently discussed. Example: Should blinding in the studies in question be seen as an important criterion? The two arguments presented were contradictory. How can confounding factors influence the results of cohort studies?
- Would it not be appropriate to join the third and fifth item of methodological limitations, given that both involve the exclusion of gray literature?
Author Response
Reviewer 1
Congratulations for the hard work. I suggest some minor revision
We thank the reviewer for the compliments.
Comment #1. Update the PROSPERO record, as important methodological changes were made to the initial proposal
Response
We thank the reviewer for this suggestion. We have updated the PROSPERO record but, there is a delay at PROSPERO. I have contacted PROSPERO and they told me that due to the covid-19 situation they are working from home what causes delay (“Your record has been submitted and an email acknowledgment sent to you. Given the long waiting time, we are receiving many emails enquiring about progress”).
Comment #2. Include the acronym PCC (Population, Concept, and Context) after your study question
Response:
We reworded this in our manuscript by introducing the PCC before the research question: This scoping review will have a broader “scope” with correspondingly less restrictive inclusion criteria. Peters et al. [32] suggests to follow the PCC (Population, Concept and Context) elements. Therefore, the following question based upon the inclusion criteria may be posed: “Which oral health interventions aiming to improve oral health in patients with a mental health disorder are described in existing literature?”
Comment # 3. In the Flow diagram, check the number of Records screened for identification of title and abstract (is it the same number found after duplicates removed?)
Response
We thank the reviewer for this suggestion. We received 2081 records through database searching and 15 records through other sources. After removing all duplicated we had 1313 records left. These 1313 records were screened for title and abstract. This is correct in the flowchart and consistent in the manuscript (line 150).
Comment # 4. Conduct a more in-depth discussion of the quality analysis, as the problems were raised, but they were not always sufficiently discussed. Example: Should blinding in the studies in question be seen as an important criterion? The two arguments presented were contradictory. How can confounding factors influence the results of cohort studies?
Response
Thank you for pointing this out. We rewrote the paragraph quality of included studies in de discussion.
Interventions combining behavioural and educational elements show to be effective in patients with SMI (diagnosis not further specified), psychotic disorder, and mood disorder. Of these studies, the methodological quality was good (n=3) [47–49]. In one RCT, the methodological quality was moderate due to insufficient follow up data [46]. In three RCT’s, there was no blinding of participants and outcome assessors. Blinding is a measure in RCT’s to reduce detection and performance bias and is an important measure in RCT’s. There is evidence that lack of blinding leads to overestimated treatment effects. If participants are not blinded, knowledge of group assignment may affect participants behaviour in an RCT [59]. This means that the treatment effects in included RCT’s [46,48,60] can be overestimated. Blinding outcome assessors can be used in order to minimise distortion in the results of the study [37]. Detection bias can arise if the knowledge of patient assignment influences the assessment of outcome measurements. This detection bias can be avoided by the blinding of those assessing outcomes in an RCT [59,61]. For included RCT’s it is not known if knowledge of a patient's assignment had influenced outcome measurements.
Quasi-experimental studies show the effectiveness in interventions combining educational and physical elements in patients with a psychotic disorder, personality disorder, anxiety disorder, mood disorder and autism spectrum disorder [28]. Educational interventions show to be effective in patients with SMI (diagnosis not further specified), eating disorder and substance abuse disorder [50,51,53,54]. Physical interventions show to be effective only in patients with autism spectrum disorder [52]. The methodological quality of the quasi-experimental studies was sufficient. But, the pretest-posttest design of the studies did not aim to compare an intervention group with a control group. The addition of control groups and sensitivity analyses can support the hypothesis that the intervention is causally associated with the outcome [62]. One 2x2 quasi-experimental study met all the requirements of the JBI checklist, however this study of Singhal et al. [54] lacked the determination of the effect of the calendar. It is not known if there was a Hawthorne effect and if the calendar was a confounder for other independent variables [54]. Furthermore, SMI has to be specified because every patient group has its own needs in managing oral health problems. Additionally, physical interventions should also be examined in other mental health patient groups beside ASD.
The quality of the cohort study (educational intervention [55]) was insufficient as confounders were not clearly identified and no strategies dealing with confounding factors were described [36]. In literature, confounding has been described as a confusion of effects [59]. To draw appropriate conclusions about the effect of the educational intervention on an outcome, the causal effects should be separated from that of the other factors that affect the outcome (e.g., age) [63]. Strategies (e.g. matching, randomization, stratification) were not used in this cohort study [64]. Due to the lack of controlling for confounding factors in included cohort study [55] it is not clear whether the conclusions were drawn appropriately or that there were other factors that affect the outcome measurements.
Comment # 5. Would it not be appropriate to join the third and fifth item of methodological limitations, given that both involve the exclusion of gray literature?
Response
We reworded the third item of methodological limitations: “Thirdly, as is the case in every review, it was possible that negative results regarding oral health interventions in patients with a mental health disorder are missed due to publication bias (e.g., exclusion of abstracts for conferences or study protocols). Additionally, due to the exclusion of grey literature, it was possible that we missed interventions that are described but not published yet. This may have affected the results and overall conclusions of this study. However, to make our scoping review and the critical appraisal more feasible for clinical practice, we decided to include peer-reviewed articles from electronic databases. Further adding to this issue is that many grey literature contain information that is not publicly readable or available”.
We deleted the fifth item in the methodological limitations.
We believe that this paper was greatly enhanced after the revisions. We would like to thank the reviewer for the feedback, which enabled us to improve the quality of the manuscript, and we do hope that our revised manuscript is now suitable for publication.
Yours sincerely,
Also on behalf of the co-authors (Nynke Boonstra, Linda Kronenberg, Annette Keuning, Stynke Castelein)
Sonja Kuipers
Reviewer 2 Report
Thank you very much for returning the revised manuscript to me, from the comments I had made in the first session the authors have edited the manuscript satisfactorily so that it can be published
Author Response
Reviewer 2
Thank you very much for returning the revised manuscript to me, from the comments I had made in the first session the authors have edited the manuscript satisfactorily so that it can be published
We thank the reviewer for the compliments.
Yours sincerely,
Also on behalf of the co-authors (Nynke Boonstra, Linda Kronenberg, Annette Keuning, Stynke Castelein)
Sonja Kuipers
Reviewer 3 Report
According to the review carried out, I consider the subject to be of interest and the methodology used is correct. However, I have same considerations that I hope will be resolved:
1) The words in table number 1 are in red.
2) The conclusions of the manuscript are very extensive.
3) Authors should describe the bibliographical references according to the journal's guidelines.
Author Response
Reviewer 2
Thank you very much for returning the revised manuscript to me, from the comments I had made in the first session the authors have edited the manuscript satisfactorily so that it can be published
We thank the reviewer for the compliments.
Yours sincerely,
Also on behalf of the co-authors (Nynke Boonstra, Linda Kronenberg, Annette Keuning, Stynke Castelein)
Sonja Kuipers
This manuscript is a resubmission of an earlier submission. The following is a list of the peer review reports and author responses from that submission.
Round 1
Reviewer 1 Report
The manuscript is globally well designed and written, however the discussion could be slightly improved in order to explain better the differences between the treatments and on the other hand the different mental disorders.
Another point is that, in my opinion the second paragraph of the introduction would fit better on the discussion topic.
Author Response
Comment #1
The manuscript is globally well designed and written, however the discussion could be slightly improved in order to explain better the differences between the treatments and on the other hand the different mental disorders.
Response
We thank the reviewer for the compliments. We have described in more detail the differences between the interventions in the discussion section: reflection on types of interventions and the quality of included studies (please see below the text in red). Next, we have made some adaptations in table 1 and in the main text (in red) to give more insight in the detected differences.
Interventions combining behavioural and educational elements show to be effective in patients with SMI (diagnosis not further specified), psychotic disorder, and mood disorder. Of these studies, the methodological quality was good (n=3) (1–3). In one RCT, the methodological quality was moderate due to insufficient follow up data (4). In three RCT’s, there was no blinding of participants and outcome assessors. Blinding outcome assessors can be used in order to minimise distortion in the results of the study (5). In these RCT’s, there is no measured medical intervention. In an RCT, when measuring social interventions, it is not reasonable to blind participants or assessors to treatment assignment or to deliver an intervention blind to treatment assignment.
Quasi-experimental studies show the effectiveness in interventions combining educational and physical elements in patients with a psychotic disorder, personality disorder, anxiety disorder, mood disorder and autism spectrum disorder (6). Educational interventions show to be effective in patients with SMI (diagnosis not further specified), eating disorder and substance abuse disorder (7–10). Physical interventions show to be effective only in patients with autism spectrum disorder (11). The methodological quality of the quasi-experimental studies was sufficient. The pretest-posttest design of the studies did not aim to compare an intervention group with a control group. Therefore, there is no evidence for causality. One 2x2 quasi-experimental study met all the requirements of the JBI checklist, however this study of Singhal et al. (10) lacked the determination of the effect of the calendar. It is not known if there was a Hawthorne effect and if the calendar was a confounder for other independent variables (10).
Furthermore, SMI has to be specified because every patient group has its own needs in managing oral health problems. Additionally, physical interventions should also be examined in other mental health patient groups beside ASD.
The quality of the cohort study (educational intervention (13)) was insufficient as confounders were not clearly identified and no strategies dealing with confounding factors were described (14).
Comment #2
Another point is that, in my opinion the second paragraph of the introduction would fit better on the discussion topic.
Response
Thank you for this suggestion. We rewrote the introduction section and added this paragraph to the discussion.
We believe that this paper was greatly enhanced after the major revisions. We would like to thank the reviewer for the feedback, which enabled us to improve the quality of the manuscript, and we do hope that our revised manuscript is now suitable for publication.
Yours sincerely,
Also on behalf of the co-authors (Nynke Boonstra, Linda Kronenberg, Annette Keuning, Stynke Castelein)
Sonja Kuipers
Reviewer 2 Report
The big question is: why did the authors choose to do a scope review and not a systematic review of intervention studies without meta-analysis (due to the heterogeneity of the studies)?
ABSTRACT
- Lines 28 and 29 - Reorganize the types of oral health intervention in ascending order, as it facilitates the reading of the text: 1) educational intervention 2) physical intervention 3) interventions combining behavioral and educational elements; 4) interventions combining educational and physical elements.
- Lines 30 and 31 - write directly that the included studies evaluated only short-term interventions (<12 months). What was the evaluation period for the only article that was not mentioned?
- Cite the quality analysis tool in the methodology.
INTRODUCTION
- The introduction confuses the reader as to the focus of the study, it also needs to be based on a more consistent and relevant literature, requiring severe changes.
- I think it is important to define what would be "mental health disorder", for the reader to understand since the beginning the patient profile of this review (because the management of a patient with depression is different from that of a patient with severe cognitive problems and limiting cognitive disability).
- Lines 42 and 43 - Review the references that support the phrase "Oral health is important for all people, particularly for vulnerable people who are exposed to oral health risk factors, e.g. patients with a mental health disorder [1,2]." It would be more appropriate to supplement with stronger references on the theme.
- Line 45: The author makes a dangerous statement in the text, supported only by a reference: "Poor oral health heightens the risk for chronic health disorders, g. Diabetes, high blood pressure, respiratory disease, and coronary heart disease." I suggest to reformulate the sentence, since the literature has not reached a consensus on the subject (eg: see article: https://academic.oup.com/eurjpc/article/25/6/598/5926482?login=true) , or to support this phrase with a well-designed systematic literature review. Maybe this phrase is unnecessary if the author chose to speak directly from patients with mental illness. I think that the central point of the introduction is to show to the reader all the risk factors that these patients are exposed (including the influence of medication use on dental caries).
- After finishing the reading of the introduction, I was with the doubt, the objective of this study is patients with mental ilness in general or just several mental illness.
- After finishing the reading of the introduction, I was with a doubt: the objective of this study is patients with any mental ilness or just several mental illness? To speak more emphatically that there is not still gold standard that guide interventions in these patients (does not exist?). You base a paragraph with a guideline from Holland. Are there not guidelines that are used globally or from other countries that can give more strength to the paragraph?
- . Line 56 - What does FEP mean?
METHODS
All of the comments below are based on the guideline (JBI Scoping Reviews) cited in the abstract.
- According to the reference that led the writing of this review, a scope review may include several types of study (The source of information can include any existing literature e.g. primary research studies, systematic reviews, meta-analyses, letters, guidelines, etc. should be explained). You need to justify why you did not include other types of study.
- Why didn't the author include guidelines? Because guidelines can contribute a lot to the discussion of the topic. This review is much more like a systematic review without meta-analysis than a scope review. Could the authors explain more about the methodological decision made?
- As explained in the abstract, make clear the reference that guided the structure of this review.
- Which acronym guided the study question? (JBI recommends the CCP)
- I did not find Appendix 1.
- The stage 2 is poorly described in the text. The abstract contains more information than the methods.
- This inclusion criterion "peer-reviewed full-text studies published in the English language were included." does not match the one described in the abstract.In the abstract was reported that the gray literature was accessed (however it was not specified in methods). Remember that the JBI recommends the inclusion of unpublished literature.
- I reinforce my consideration about the exclusion of guidelines: Is your review really a scope review?
- Explain how the language restriction can limit your review.
- The term "psychiatric disorder" first appears on line 122. Standardize the terminologies used throughout the text.
- After reading the methodology, I still had doubts about the profile of the patients included in this review.
- Has a reference manager been used to assist reviewers in the study selection process?
- On the description of the quality assessment, I believe that the citation of questions as an example are unnecessary
RESULTS
- The authors mention Prisma only in the results, without referencing
- Line 177 to 181. Were 11 or 9 studies included in this review?
- The difference between the data in Outcome and "Outcome measurements on oral health" was confused in table 2
- The categories presented in the abstract should be presented in Table 2 (1. an educational intervention 2. physical intervention 3. interventions combining behavioral and educational elements; 4. interventions combining educational and physical elements)
- If, after reflection, the authors define that this study is a systematic review, I suggest working better on the results of the quality assessment (also include in the discussion).
- Line 270: Shouldn't the loss of follow-up from the Addams study have an impact on methodological quality? (It is green in "Complete follow-up or differences analyzed properly")
DISCUSSION
- Line 275. To reflect and include in the discussion: Is this reminder system a method that would be easily implemented?
- Most studies included institutionalized patients. What would be the impact of these interventions outside the centers studied? Shouldn't this be considered in the title, objectives and / or discussion of this review?
Author Response
Comment #1
The big question is: why did the authors choose to do a scope review and not a systematic review of intervention studies without meta-analysis (due to the heterogeneity of the studies)?
Response
Indeed, we discussed this consideration prior to the study. Munn et al. (2018) published an interesting paper which guides authors to choose between a scoping or a systematic review. This paper supports our choice to choose for a scoping review. Please find below our arguments.
Our research aims to provide a broad overview of oral health interventions for patients with a mental health disorder and to evaluate the study quality of included studies. Until now, there are just a few studies published about oral health in mental health disorders. Therefore, our first step was to write a scoping review which would address the topic as broad as possible. What kind of interventions are described? Which study designs are used? Which outcomes and measurement instruments are used? What effects are found? And what about the study quality of the included - quantitative and qualitative - studies?
We would have chosen for a systematic review (or even better for a meta-analysis) if many RCT’s would have been published about oral health in mental health disorders. Systematic reviews might typically focus on a well-defined research question in advance. For instance: “What is the effect of an educational oral health intervention in patients with an autism spectrum disorder?” In this case, study designs are also identified in advance (Wright et al. (2007).
Although, we chose for a scoping review to give a broad overview of oral health interventions (and other meaningful outcomes for clinical practice) in mental health disorders, we were interested in the study quality of the included studies. This way, we could give some insight in the (potential) effectiveness of oral health interventions in the future. Several critical appraisal tools of the Joanna Briggs Institute were used.
We understand that the reviewer suggested a systematic review as we critically appraised the literature and also summarized the effects in the included studies, but only four RCTs could be included within different diagnostic patients groups, different types of interventions, and different outcome. The heterogeneity was very high.
We added the following sentence in the methods section. “This research aims to provide a broad overview of oral health interventions for patients with a mental health disorder and to evaluate the study quality of included studies. Therefore, a systematic scoping review with a critical appraisal was conducted using the Joanna Briggs Institute (JBI) methodology for scoping reviews and their checklists (32–36). A scoping review seeks to provide thorough coverage of literature and is thereby a mechanism for findings for mental health professionals (32). In contrast to a systematic review, a scoping review adopts more flexibility in study selection, e.g. more flexibility with inclusion and exclusion criteria and the search terms may be redefined during the process and more criteria can be devised post hoc (32).”
ABSTRACT
Comment #2
- Lines 28 and 29 - Reorganize the types of oral health intervention in ascending order, as it facilitates the reading of the text: 1) educational intervention 2) physical intervention 3) interventions combining behavioral and educational elements; 4) interventions combining educational and physical elements.
Response
We reorganised the text in the abstract and in our manuscript in the following order: I) educational interventions; II) physical interventions; III) interventions combining behavioural and educational elements and IV) interventions combining educational and physical elements.
Comment #3
- Lines 30 and 31 - write directly that the included studies evaluated only short-term interventions (<12 months). What was the evaluation period for the only article that was not mentioned?
Response
Thank you for this suggestion. We reworded this in our abstract and in our discussion: “All studies (n=11) had an evaluation period ≤12 months. Nine studies showed an effect on the short term (≤12 months) with regard to oral health knowledge, oral health behaviour, or physical oral health outcomes (e.g. plaque index). Two studies showed no effects on any outcome.”
Comment #4
- Cite the quality analysis tool in the methodology.
Response
We added the tool in our abstract and in our manuscript: “Therefore, a systematic scoping review with a critical appraisal was conducted using the Joanna Briggs Institute (JBI) methodology for scoping reviews and their checklists.”
In the methods section we added the following sentences:
On page 3: Therefore, a systematic scoping review with a critical appraisal was conducted using the Joanna Briggs Institute (JBI) methodology for scoping reviews (32,33) and their checklists (34–36).
On page 4: To evaluate the methodological quality, all studies were critically appraised using the checklists developed by the Joanna Brigg’s Institute (JBI). RCT’s were assessed with the JBI-tool developed for RCT’s consisting of 13 items (36). Non-randomised intervention studies and pretest-posttest studies were assessed with the JBI-tool for Quasi-Experimental Studies (9 items) (34). Cohort studies were appraised with the 11-item JBI-tool developed for cohort studies (35). Two researchers (S.K. and A.K.) critically assessed the methodological quality independently. Disagreements were discussed. Cohen’s Kappa statistics were calculated to test inter-rater reliability (42).
INTRODUCTION
Comment #5
- The introduction confuses the reader as to the focus of the study, it also needs to be based on a more consistent and relevant literature, requiring severe changes.
Response
Thank you for this feedback. We rewrote the introduction as follows:
The World Health Organisation (WHO) emphasises that oral health is integral and essential to general health and wellbeing (1,2). Oral health is improved in the general population, but vulnerable patients (e.g. patients diagnosed with a mental health disorder) have not benefited of the worldwide improvement in oral health (3). Poor oral health is associated with diabetes (both type 1 and 2) (4), respiratory disease and abdominal obesity (4–6). It might also be related to cardiovascular diseases (4,7), but cigarette smoking might influence this relationship (8).
Nearly 20% of the population worldwide suffers from a mental health disorder (9,10), this outlines the importance of oral health in patients diagnosed with a mental health disorder (Cormac & Jenkins (11)) who are exposed to more oral health risk factors (3,11–14).
Several risk factors to poor oral health in patients with a mental health disorder were described (15). Many patients consume medication such as antipsychotics, antidepressants, and lithium. A dry mouth (21, 22) is a side-effect of the medication which can increase plaque (16–19). Next, oral health will be worsened by the consumption of sugary sweets and sugary drinks (20) which are more frequently used in patients with a mental health disorder. Inadequate oral health self-management, a lower tooth brushing frequency, a lack of motivation for proper oral hygiene and health care habits and poor psychosocial functioning are known as other barriers for adequate oral health in patients diagnosed with a mental health disorder (15,21–23). Bad breath (halitosis) may lead to poor self-image, low self-esteem, decreased self-confidence, social phobia, loneliness, depression and suicidal intents in the general population (24,25).
Thus, poor oral health affects quality of life and daily functioning in the general population and especially in patients with a mental health disorder (1,2). As a consequence, patients living with severe mental illness (SMI) (e.g. schizophrenia or related psychotic disorders, bipolar disorder,) are almost three times more likely to have lost all of their teeth compared to the general population (26).
It is evident that routine and effective oral care is necessary for maintaining oral health of in- and outpatients (27). Mental health professionals (e.g., nurses) have an important role in the care for (out)patients with a mental health disorder. Therefore, nurses should consider oral health care as an essential part of their care for patients with mental health disorders (28).
Until now, existing NICE-guidelines primarily focus on oral health in general practice (29) and on adults in care homes (30). No NICE-guideline focusses on oral health interventions of patients diagnosed with a mental health disorder, their oral health needs and risk factors (e.g., the use of antipsychotic medication). A British guideline titled “Oral Health Care for People with Mental Health Problems” (22) describes the severity and prevalence of oral health problems in mental health. This guideline does not meet the needs with outdated literature. The evidence of interventions is mostly focussed on institutionalised elderly and not on patients with mental health disorders. It is important to outline interventions in groups of mental health disorder due to the differences in management (e.g., the management of oral health of a patient with depression might differ from that of a patient with severe cognitive problems).
Considering the poor oral health, increased risk factors, the high burden of poor oral health (11–13,26,31) and the lack of interventions in existing guidelines, it is important to explore which oral health interventions are available for our patient population in existing research. The primarily research question is: “Which oral health interventions aiming to improve oral health in patients with a mental health disorder are described in existing literature?”
We aim to provide a broad overview of oral health interventions for patients with a mental health disorder including an evaluation of the study quality.
Comment #6
- I think it is important to define what would be "mental health disorder", for the reader to understand since the beginning the patient profile of this review (because the management of a patient with depression is different from that of a patient with severe cognitive problems and limiting cognitive disability).
Response
Our study aimed to identify interventions in people with a mental health disorder. Their main diagnosis is in accordance with the DSM-IV or 5 (since 2013) (41). The following diagnoses belong to SMI: psychotic disorders, mood disorders, anxiety disorders, autism spectrum disorders, eating disorders, and substance abuse disorders. Studies which primarily focused on dementia and mental health retardation were excluded. Mental health disorder is now clearly defined in the materials and methods section (stage 1).
In the results, we describe our findings per patient diagnosis group. If we would focus on symptoms (for instance, cognitive problems as suggested by the reviewer), the aim of the scoping review would completely change. We were primarily interested in an overview of oral health interventions in people with a mental health disorder.
Severe mental illness (SMI) refers to people with psychological problems that are often so debilitating that their ability to engage in functional and occupational activities is severely impaired. In literature, SMI is a container term for patients and often not further specified in e.g. schizophrenia or related psychotic disorders, depression (Delespaul, 2013).
- American Psychiatric Association. Diagnostic and statistical manual of mental disorders (DSM-5). American Psychiatric Pub; 2013.
Comment #7
- Lines 42 and 43 - Review the references that support the phrase "Oral health is important for all people, particularly for vulnerable people who are exposed to oral health risk factors, e.g. patients with a mental health disorder [1,2]." It would be more appropriate to supplement with stronger references on the theme.
Response
We rewrote the introduction, please see our answer on comment 5.
We rewrote this specific sentence and added three references: “Thus, poor oral health affects quality of life and daily functioning in the general population and especially in patients with a mental health disorder (1,2). As a consequence, patients living with severe mental illness (SMI) (e.g. schizophrenia or related psychotic disorders, bipolar disorder,) are almost three times more likely to have lost all of their teeth compared to the general population (26).”
Comment #8
- Line 45: The author makes a dangerous statement in the text, supported only by a reference: "Poor oral health heightens the risk for chronic health disorders, e.g. diabetes, high blood pressure, respiratory disease, and coronary heart disease." I suggest to reformulate the sentence, since the literature has not reached a consensus on the subject (eg: see article: https://academic.oup.com/eurjpc/article/25/6/598/5926482?login=true), or to support this phrase with a well-designed systematic literature review. Maybe this phrase is unnecessary if the author chose to speak directly from patients with mental illness. I think that the central point of the introduction is to show to the reader all the risk factors that these patients are exposed (including the influence of medication use on dental caries).
Response
Thank you for this suggestion. This is reworded in our rewritten introduction: “Poor oral health is associated with diabetes (both type 1 and 2) (4), respiratory disease and abdominal obesity (4–6). It might also be related to cardiovascular diseases (4,7), but cigarette smoking might influence this relationship (8).”
Comment #9
- After finishing the reading of the introduction, I was with the doubt, the objective of this study is patients with mental illness in general or just several mental illness?
Response
We rewrote the introduction to make this more clear. To give a direct answer; we are interested in patients with a mental health disorder. Their main diagnosis is in accordance with the DSM-IV or 5 (since 2013) (American Psychiatric Association, 2013). The following diagnoses belong to SMI: psychotic disorders, mood disorders, anxiety disorders, autism spectrum disorders, eating disorders, and substance abuse disorders.
Please see also our answer on comment 6.
Comment #10
- To speak more emphatically that there is not still gold standard that guide interventions in these patients (does not exist?). You base a paragraph with a guideline from Holland. Are there not guidelines that are used globally or from other countries that can give more strength to the paragraph?
Response
Thank you for your suggestion. We deleted this sentence. This is reworded in the introduction;
Until now, existing NICE-guidelines primarily focus on oral health in general practice (29) and on adults in care homes (30). No NICE-guideline focusses on oral health interventions of patients diagnosed with a mental health disorder, their oral health needs and risk factors (e.g., the use of antipsychotic medication). A British guideline titled “Oral Health Care for People with Mental Health Problems” (22) describes the severity and prevalence of oral health problems in mental health. This guideline does not meet the needs with outdated literature. The evidence of interventions is mostly focussed on institutionalised elderly and not on patients with mental health disorders. It is important to outline interventions in groups of mental health disorder due to the differences in management (e.g., the management of oral health of a patient with depression might differ from that of a patient with severe cognitive problems).
Based on the results of current study we reworded this sentence: Due to the heterogeneity in both interventions, diagnostic groups and outcomes, one golden standard oral health intervention cannot be advised yet, although the methodological quality of studies seems sufficient.
Comment #11
- . Line 56 - What does FEP mean?
Response
FEP means first episode psychosis. We reworded this in the manuscript: “(e.g. first episode psychosis (FEP))”
METHODS
All of the comments below are based on the guideline (JBI Scoping Reviews) cited in the abstract.
Comment #12
- According to the reference that led the writing of this review, a scope review may include several types of study (The source of information can include any existing literature e.g. primary research studies, systematic reviews, meta-analyses, letters, guidelines, etc. should be explained). You need to justify why you did not include other types of study.
Response
We reworded this in our manuscript (stage 2): “For this study, we included peer-reviewed full-text studies published in the English language. Randomised controlled trials (RCT’s), non-randomised intervention studies, observational studies (cohort, case-control and cross-sectional studies), and qualitative studies about oral health interventions in patients with a mental health disorder were included. Systematic reviews and meta-analyses are included and cross-referencing was applied to search for other relevant articles. Grey literature and guidelines were excluded, because they are composed for knowledge artefacts and were not peer-reviewed (41).”
Please see also our answer on comment 1.
Comment #13
- Why didn't the author include guidelines? Because guidelines can contribute a lot to the discussion of the topic. This review is much more like a systematic review without meta-analysis than a scope review. Could the authors explain more about the methodological decision made?
Response
Please see also our answer on comment 1 and 12.
Comment #14
- As explained in the abstract, make clear the reference that guided the structure of this review.
Response
We reworded this in the methods section of our manuscript. “Therefore, a systematic scoping review with a critical appraisal was conducted using the Joanna Briggs Institute (JBI) methodology for scoping reviews (32,33) and their checklists (34–36).”
Comment #15
- Which acronym guided the study question? (JBI recommends the CCP).
Response
We suppose that the reviewer means PCC (Population, Concept, and Context) instead of CCP. The primary research question of this scoping review was based on JBI PCC-method (Peters, 2015) and reworded in our manuscript: “Which oral health interventions aiming to improve oral health in patients with a mental health disorder are described in existing literature?”
Comment #16
- I did not find Appendix 1.
Response
We added the ‘Search strategy’ as Appendix 1 to our manuscript.
Comment #17
- The stage 2 is poorly described in the text. The abstract contains more information than the methods.
Response
We added more information in stage 2: Identifying Relevant Studies:
A search strategy was developed in collaboration with a Medical Information Officer (TI) from the University Medical Centre of Groningen (The Netherlands). The search strategy was conducted from the research question and the definitions (Appendix 1). Three electronic databases (MEDLINE, CINAHL and PsycINFO) were searched from their inception until December 2020 and cross-referencing was applied. RefWorks Version 2 was used in the study selection process.
For this study, we included peer-reviewed full-text studies published in English. Randomised controlled trials (RCT’s), non-randomised intervention studies, observational studies (cohort, case-control and cross-sectional studies), and qualitative studies about oral health interventions in patients with a mental health disorder. Systematic reviews and meta-analyses are included and cross-referencing was applied to search for additional relevant articles. Grey literature and guidelines were excluded, because they are composed for knowledge artefacts that are not the product of a peer-reviewed process (41).
Exclusion criteria were: i) no focus on an oral health intervention; ii) an exclusive focus on exploring the severity of oral health problems; iii) absence of explicit reference to mental health disorders; iv) primary focus on dementia or mental health retardation; v) interventions focusing on the frequency of appointments with the dentist.
The search identified 1313 potential papers after removing duplicates (figure 1). Two researchers (SK & AK) screened the abstracts on eligibility based on title and abstract.
Comment #18
- This inclusion criterion "peer-reviewed full-text studies published in the English language were included." does not match the one described in the abstract. In the abstract was reported that the gray literature was accessed (however it was not specified in methods). Remember that the JBI recommends the inclusion of unpublished literature.
Response
Although we were aware that JBI recommends the inclusion of unpublished literature and grey literature, we decided to include only peer-reviewed studies to assure a certain quality of the included studies. We deleted grey literature of our abstract. We added to our manuscript that we excluded grey literature and guidelines.
Please see also our answer on comment 1 and 12.
Furthermore, we discussed this in the limitations, see also comment 20
Comment #19
- I reinforce my consideration about the exclusion of guidelines: Is your review really a scope review?
Response
Please see our response on comment 1.
Comment #20
- Explain how the language restriction can limit your review.
Response
We discussed language restriction and the exclusion of grey literature and guidelines.
We discussed this in our research group and reworded this in our manuscript (limitations):
Fourthly, due to the limitation of publications in English, it is possible that we missed peer-reviewed studies on oral health interventions in other languages. However, it is unknown if this affected the results and overall conclusions of this study. Fifthly, due to the exclusion of grey literature, it was possible that we missed interventions who are described but not published yet. This may have affected the results and overall conclusions of this study. However, to make our scoping review and the critical appraisal more feasible for clinical practice, we decided to include peer-reviewed articles from electronic databases. Further adding to this issue is that many grey literature contain information that not publicly readable or available.
Comment #21
- The term "psychiatric disorder" first appears on line 122. Standardize the terminologies used throughout the text.
Response
We changed the term “psychiatric disorder” into: mental health disorder.
Comment #22
- After reading the methodology, I still had doubts about the profile of the patients included in this review.
Response
Please see our response on comment 6.
Comment #23
- Has a reference manager been used to assist reviewers in the study selection process?
Response
We added the following on our manuscript: “RefWorks Version 2 was used in the study selection process.”
Comment #24
- On the description of the quality assessment, I believe that the citation of questions as an example are unnecessary.
Response
We agree with the reviewer. We deleted the questions of our manuscript.
RESULTS
Comment #25
- The authors mention Prisma only in the results, without referencing
Response
We added two references to our manuscript:
- The Joanna Briggs Institute. The Joanna Briggs Institute Reviewers’ Manual 2015: Methodology for JBI scoping reviews. Joanne Briggs Inst [Internet]. 2015;(August):1–24. Available from: http://joannabriggs.org/assets/docs/sumari/ReviewersManual_Mixed-Methods-Review-Methods-2014-ch1.pdf
- Moher D, Liberati A, Tetzlaff J, Altman DG, Altman D, Antes G, et al. Preferred reporting items for systematic reviews and meta-analyses: The PRISMA statement. PLoS Med. 2009;6(7).
Additionally, in the diagram, we did not mention the cross-references. We added this to diagram
Comment #26
- Line 177 to 181. Were 11 or 9 studies included in this review?
Response
Eleven studies were included, we deleted the double sentence.
Comment #27
- The difference between the data in Outcome and "Outcome measurements on oral health" was confused in table 2.
Response
We reworded “Outcome measurements on oral health” into table 2: Measurement instrument. This is also reworded in our manuscript.
Comment #28
- The categories presented in the abstract should be presented in Table 2 (1. educational intervention 2. physical intervention 3. interventions combining behavioural and educational elements; 4. interventions combining educational and physical elements)
Response
We added a column with “Type of oral health interventions” in Table 2.
Comment #29
- If, after reflection, the authors define that this study is a systematic review, I suggest working better on the results of the quality assessment (also include in the discussion).
Response
Thank you for your feedback. We discussed this in our research group. Based on the research question and the aim of this study, a scoping review was more appropriate. We agree with the reviewer that the results of the quality assessment could be described in more detail. We reworded this in the discussion in the section: quality of included studies.
Interventions combining behavioural and educational elements show to be effective in patients with SMI (diagnosis not further specified), psychotic disorder, and mood disorder. Of these studies, the methodological quality was good (n=3) (1–3). In one RCT, the methodological quality was moderate due to insufficient follow up data (4). In three RCT’s, there was no blinding of participants and outcome assessors. Blinding outcome assessors can be used in order to minimise distortion in the results of the study (5). In these RCT’s, there is no measured medical intervention. In an RCT, when measuring social interventions, it is not reasonable to blind participants or assessors to treatment assignment or to deliver an intervention blind to treatment assignment.
Quasi-experimental studies show the effectiveness in interventions combining educational and physical elements in patients with a psychotic disorder, personality disorder, anxiety disorder, mood disorder and autism spectrum disorder (6). Educational interventions show to be effective in patients with SMI (diagnosis not further specified), eating disorder and substance abuse disorder (7–10). Physical interventions show to be effective only in patients with autism spectrum disorder (11). The methodological quality of the quasi-experimental studies was sufficient. The pretest-posttest design of the studies did not aim to compare an intervention group with a control group. Therefore, there is no evidence for causality. One 2x2 quasi-experimental study met all the requirements of the JBI checklist, however this study of Singhal et al. (10) lacked the determination of the effect of the calendar. It is not known if there was a Hawthorne effect and if the calendar was a confounder for other independent variables (10).
Furthermore, SMI has to be specified because every patient group has its own needs in managing oral health problems. Additionally, physical interventions should also be examined in other mental health patient groups beside ASD.
The quality of the cohort study (educational intervention (13)) was insufficient as confounders were not clearly identified and no strategies dealing with confounding factors were described (14).
Comment #30
- Line 270: Shouldn't the loss of follow-up from the Addams study have an impact on methodological quality? (It is green in "Complete follow-up or differences analyzed properly")
Response
We agree with the reviewer. We discussed this in our research group and changed this in table 4. It is also reworded in our manuscript: “Besides double blindness, three RCTs met all the other JBI criteria (1–3). Additionally, one RCT could also not complete its follow up assessment (4).”
DISCUSSION
Comment #31
- Line 275. To reflect and include in the discussion: Is this reminder system a method that would be easily implemented?
Response
We included this in our discussion (reflection on types of interventions p. 26): “Reminder strategies combined with oral health education showed to have a significant effect on behaviour of patients with a mental health disorder (schizophrenia, depression, bipolar disorder) (46). Reminder systems, such as post-it, are easy to implement. Alqahtani et al. (56) show that reminder strategies enable a system to remind the user to perform the target behaviour. Reminders are often implemented to remind users to perform activity in mental health disorders (56). There are no studies examining the effects of reminder strategies focusing on oral health in mental health apps. Therefore, further research on reminder strategies improving oral health is needed”.
- Most studies included institutionalized patients. What would be the impact of these interventions outside the centers studied? Shouldn't this be considered in the title, objectives and / or discussion of this review?
Response
Our manuscript did not aim to study a specific kind of setting of inpatients or outpatients. Therefore, we did not include the setting in the title. However, we agree with the reviewer that it is important to describe the setting of the target population, therefore, we added it in 3.4 and in the discussion.
In the manuscript, we described this in 3.4 synthesis (P.19): “These studies included outpatients (n=7) or inpatients (n=4).”
In the discussion we added: “Of all studies, seven studies were focussed on outpatients, four studies were focussed on inpatients. Looking for interventions, it is important to look carefully whether interventions are developed for inpatients or outpatients, because the findings may not be generalizable to all patients with a mental health disorder.”
We believe that this paper was greatly enhanced after the major revisions. We would like to thank the reviewer for the feedback, which enabled us to improve the quality of the manuscript, and we do hope that our revised manuscript is now suitable for publication.
Yours sincerely,
Also on behalf of the co-authors (Nynke Boonstra, Linda Kronenberg, Annette Keuning, Stynke Castelein)
Sonja Kuipers
Reviewer 3 Report
Manuscript in line with the criteria for a review, special needs patients need support and a theorized plan.
In this regard, I would add to the discussion the training of caregivers, and the proactive approach for clinical and home management through the use of electric toothbrushes and the inclusion of probiotics during home oral hygiene methods in order to maintain homeostasis at home, internal oral cavity and a balanced microbiome.I would add some references
I would arrange so that you can better read Tables 1 and 4, very confusing, evaluations not aligned with the text.
Author Response
Manuscript in line with the criteria for a review, special needs patients need support and a theorized plan.
We thank the reviewer for the compliments.
Comment #1
In this regard, I would add to the discussion the training of caregivers, and the proactive approach for clinical and home management through the use of electric toothbrushes and the inclusion of probiotics during home oral hygiene methods in order to maintain homeostasis at home, internal oral cavity and a balanced microbiome. I would add some references
Response
Thank you for this suggestion. We added this in our manuscript in the discussion (P.28): “There are constantly new insights regarding oral health. A recent study for example showed the effectiveness of a mechanical and ultrasonic toothbrush on oral biofilm removal (58). This highlights the proactive approach for clinical and home management through the use of mechanical or ultrasonic toothbrushes in outpatients and inpatients with a mental health disorder. Furthermore, a recent study on students of Lee et al. (59) showed that ingestion of the oral probiotic Weissella cibaria can help reduce subjective halitosis and improve oral-health-related quality of life. However, this was not tested as intervention in patients with a mental health disorder. Therefore, further research on the use of oral probiotic Weissella cibaria could be interesting.”
Comment #2
I would arrange so that you can better read Tables 1 and 4, very confusing, evaluations not aligned with the text.
Response
We have made some adaptations in the results and in table 1 and 4.
We believe that this paper was greatly enhanced after the major revisions. We would like to thank the reviewer for the feedback, which enabled us to improve the quality of the manuscript, and we do hope that our revised manuscript is now suitable for publication.
Yours sincerely,
Also on behalf of the co-authors (Nynke Boonstra, Linda Kronenberg, Annette Keuning, Stynke Castelein)
Sonja Kuipers